# CCN2 Activates RIPK3, NLRP3 Inflammasome, and NRF2/Oxidative Pathways Linked to Kidney Inflammation

**DOI:** 10.3390/antiox12081541

**Published:** 2023-07-31

**Authors:** Sandra Rayego-Mateos, Laura Marquez-Exposito, Pamela Basantes, Lucia Tejedor-Santamaria, Ana B. Sanz, Tri Q. Nguyen, Roel Goldschmeding, Alberto Ortiz, Marta Ruiz-Ortega

**Affiliations:** 1Cellular Biology in Renal Diseases Laboratory, IIS-Fundación Jiménez Díaz, Universidad Autónoma Madrid, 28040 Madrid, Spain; srayego@fjd.es (S.R.-M.); lauramarquezexposito@gmail.com (L.M.-E.); pamelabasantes@hotmail.com (P.B.); lucia.tejedor@quironsalud.es (L.T.-S.); 2Ricor2040, 28029 Madrid, Spain; 3Division of Nephrology and Hypertension, IIS-Fundación Jiménez Díaz-Universidad Autónoma Madrid, 28040 Madrid, Spain; asanzb@fjd.es (A.B.S.); aortiz@fjd.es (A.O.); 4Department of Pathology, University Medical Center Utrecht, H04.312, Heidelberglaan 100, 3584 Utrecht, The Netherlands; t.q.nguyen@umcutrecht.nl (T.Q.N.); r.goldschmeding@umcutrecht.nl (R.G.)

**Keywords:** CCN2, NLRP3 inflammasome, RIPK3, inflammation, NRF2, oxidation, acute kidney injury

## Abstract

Inflammation is a key characteristic of both acute and chronic kidney diseases. Preclinical data suggest the involvement of the NLRP3/Inflammasome, receptor-interacting protein kinase-3 (RIPK3), and NRF2/oxidative pathways in the regulation of kidney inflammation. Cellular communication network factor 2 (CCN2, also called CTGF in the past) is an established fibrotic biomarker and a well-known mediator of kidney damage. CCN2 was shown to be involved in kidney damage through the regulation of proinflammatory and profibrotic responses. However, to date, the potential role of the NLRP3/RIPK3/NRF2 pathways in CCN2 actions has not been evaluated. In experimental acute kidney injury induced with folic acid in mice, CCN2 deficiency diminished renal inflammatory cell infiltration (monocytes/macrophages and T lymphocytes) as well as the upregulation of proinflammatory genes and the activation of NLRP3/Inflammasome-related components and specific cytokine products, such as IL-1β. Moreover, the NRF2/oxidative pathway was deregulated. Systemic administration of CCN2 to C57BL/6 mice induced kidney immune cell infiltration and activated the NLRP3 pathway. RIPK3 deficiency diminished the CCN2-induced renal upregulation of proinflammatory mediators and prevented NLRP3 modulation. These data suggest that CCN2 plays a fundamental role in sterile inflammation and acute kidney injury by modulating the RIKP3/NLRP3/NRF2 inflammatory pathways.

## 1. Introduction

Chronic kidney disease (CKD) is emerging as one of the fastest-growing causes of death and is expected to become the fifth global cause of death by 2040 [1]. Currently, there are no treatments for acute kidney injury (AKI) or to prevent the progression of AKI into CKD [1,2,3,4]. Hence, further research is needed to understand this complex disease and to develop new therapeutic strategies that restore renal function or prevent the progression of the disease.

One common feature of CKD and AKI is the activation of the inflammatory response [2]. Preclinical data suggest that anti-inflammatory interventions could be used as therapeutic options for kidney disease. However, no therapy specifically targeting inflammation is in clinical use [1,2,3,4]. Among these targets, the canonical activation of NLRP3 inflammasome was shown to be involved in kidney diseases, mainly by the regulation of necroinflammation [5,6]. In different experimental renal diseases, the genetic or pharmacological inhibition of NLRP3 has demonstrated beneficial effects [7,8,9,10,11]. However, IL-1β blockers, which target the key inflammasome effector, failed to mitigate kidney disease [5,6]. These findings remark the need for further research on therapies targeting NLRP3 for kidney disease.

Previous studies in experimental AKI described the link between inflammation and necroptosis, a cell death form of regulated necrosis [12,13,14]. Necroptosis is initiated by the phosphorylation of receptor-interacting protein kinase-3 (RIPK3) by RIPK1 followed by RIPK3 phosphorylated mixed-lineage kinase domain-like protein (MLKL) phosphorylation, which leads to disruption of the plasma membrane [14,15,16,17,18]. However, RIPK3 can also mediate inflammation and/or fibrosis in the kidney independent of cell death, as it has been observed in different experimental models [19].

Early studies demonstrated that the matricellular protein named cellular communication network factor 2 (CCN2) acts as a growth factor or cytokine, eliciting multiple cellular responses, including proliferation, ECM synthesis, proinflammatory responses, angiogenesis, and ECM regulation [20,21,22,23,24,25,26,27]. Recent studies using conditional deficient mice described the complexity of CCN proteins and extended their role to matricellular proteins, which are key ECM components implicated in cellular communication and ECM homeostasis [28,29,30]. Many preclinical studies demonstrated that CCN2 is involved in the pathogenesis of kidney damage by promoting inflammation, fibrosis, and cellular senescence, both in vitro and in vivo [23,31,32,33]. Moreover, CCN2 may be a therapeutic target in experimental kidney diseases [34,35,36], as well as a potential biomarker for human CKD [20,37,38]. Experimental evidence suggests that CCN2 can regulate redox processes linked to proinflammatory responses, as described in vascular smooth muscle cells [39], but kidney data are scarce. In experimental AKI induced by ischemia-reperfusion injury (IRI), we have previously described that CCN2 regulates the redox/NRF2 pathway associated with the activation of cellular senescence [32,40]. Although the activation of several proinflammatory-related signaling pathways was shown to be involved in CCN2 actions in the kidney [33,34,35,36,37,38,39,40], to date, the potential role of the NLRP3/RIPK3 pathway has not been investigated. To fill this knowledge gap, we evaluated the response of CCN2 conditionally deficient mice to AKI induced with a nephrotoxic compound (folic acid administration) and also the direct effects of CCN2 administration on murine kidneys.

## 2. Materials and Methods

### 2.1. Animals

The experimental design of mice models was performed according to the European Community guidelines for animal experiments and the ARRIVE guidelines. The experiments were also developed with consent from the Experimental Animal Ethics Committee of the Comite de Etica de Experimentación Animal of the IIS-Fundación Jiménez Díaz (Proex065/18 of the Comunidad de Madrid).

Our local animal facilities had special pathogen-free conditions, and mice had free access to food, water, and normal circadian cycles. Animals were euthanized with an overdose of CO_2_. Blood and urine samples were collected, and kidneys were perfused in situ with serum saline before removal. Half of each kidney (2/4) was fixed, embedded in paraffin, and used for immunohistochemistry, and the rest was snap-frozen in liquid nitrogen for renal cortex RNA and protein studies.

#### 2.1.1. Generation of Conditional CCN2 Knockout (KO) Mice

The generation of time-conditional CCN2 complete knockout mice, homozygous CCN2 flox mice (generated by Dr. Andrew Leask’s laboratory; University of Western Ontario, Canada [41,42]) were crossed with tamoxifen-inducible Cre recombinase (CreERT2) mice under the control of the ROSA26 locus (ROSA26CreERT2; The Jackson Laboratory, Bar Harbor, ME, USA). ROSA26CreERT2 and CCN2 flox mice, both with C57Bl/6J genetic background, were crossbred for five generations, and homozygous CCN2/floxROSA26-ERT/Cre mice were used in our experimental design described in this manuscript. The genotype of mice was confirmed with polymerase chain reaction (PCR) using the following primers: CCN2 flox forward (5′-AAAGTCGCTCTGAGTTGTTAT-3′) and reverse (5′-CCTGATCCTGGCAATTTCG3′); ROSA26CreERT2 forward (5′-AATACCAATGCACTTGCCTGGATGG3′) and reverse (5′-GAAACAGCAATTACTACAACGGGAGTGG-3′) and (5′-GGAGCGGGAGAAATGGATATG-3′). CCN2 gene deletion in male mice was induced with the i.p administration of 4 injections of 10 mg/mL tamoxifen (resuspended in corn oil) over 7 days. After that, a two-week washout period (counting from the last injection) was developed, and then Ccn2 deletion was confirmed with polymerase chain reaction (PCR) using CCN2-floxed Forward: 5′-AATACCAATGCACTTGCCTGGATGG-3′ and CCN2-floxed Reverse: 5′-GAAACAGCAATTACTACAACGGGAGTGG-3′ primers. The amplified DNA was resolved by size in agarose gel (1.5%).

#### 2.1.2. Animal Models of Renal Damage

Folic acid nephropathy (FA) is a classical AKI model with tubular cell involvement characterized by acute functional renal failure, cell death, immune cell infiltration, and tubular regeneration [43]. *CCN2flox/floxROSA26-ERT/Cre* mice were injected with corn oil (vehicle) or with tamoxifen to induce CCN2 deletion. Three weeks later, CCN2 deletion was confirmed, as described in Section 2.1.1. and mice (control and CCN-deleted groups) were divided into two additional groups: one received a single FA intraperitoneal injection ((250 mg/kg; Sigma-Aldrich, St. Louis, MO, USA) in 0.3 mol/L sodium bicarbonate or vehicle (n = 4–8 animals)) and the other received vehicle (n = 4–8 animals). The mice were sacrificed 48 h later during the acute phase of AKI, characterized by an inflammatory response in the kidney. 

No renal lesions or inflammation were observed in the vehicle (corn oil) (vehicle group was considered the control) or tamoxifen+vehicle groups. In the absence of definitive evidence for an impact of sex on disease susceptibility [43], male mice were used in line with a previous study ischemia–reperfusion injury model also developed in male conditional CCN2 deficient mice [32].

CCN2 administration: Studies were performed using adult male C57BL/6 mice bred in our animal facility (9–12 weeks old, 20 g). Gender was aligned with the prior experiment, and male mice were used. C57BL/6 male (n = 4–8 mice per group) mice received a single intraperitoneal injection of 2.5 ng/g of body weight recombinant CCN2(IV) (endotoxin levels, 0.01 units Preprotech; Cranbury, NJ, USA) dissolved in saline, as previously described [23,31,33], and were studied 24 h later. RIPK3-KO mice (provided by Kim Newton and Vishva Dixit [44]; Genentech, South San Francisco, CA, USA) with a C57Bl/6 background or wildtype littermates were killed 24 h after intraperitoneal CCN2(IV) injection.

### 2.2. Protein Studies

Total protein samples were obtained from frozen kidney tissue using lysis buffer, as previously described by [45], and quantified using a BCA protein assay kit (ThermoScientific; Waltham, MA, USA). Different proteins of interest were separated using 8–15% acrylamide SDS-PAGE gels (25–50 μg of total protein loaded), as described by [45]. After electrophoresis, samples were transferred to polyvinylidenedifluoride membranes (PVDF) (Millipore; Burlington, MA, USA), blocked in TBS (plus 0.1% Tween 20 with 5% dry non-fat milk) for 1 h at room temperature, and then incubated in the same buffer with different primary antibodies overnight at 4 °C. After washing the membranes, HRP (horseradish peroxidase)-conjugated secondary antibody (Invitrogen) incubation was developed for 1 h at room temperature and revealed using an ECL kit (Amersham Biosciences; Buckinghamshire, UK). Results were analyzed/densitometered with an Amersham Imager 600 (GEHealthcare; Chicago, IL, USA) and using Quantity One software (Biorad, Hercules, CA, USA), respectively. The following primary antibodies were used (dilution): anti-MLKL (1:1000, ab196436; Abcam; Waltham, MA, USA); anti-NLRP3 (1:1000, Adipogen; San Diego, CA, USA, AG20B-0014-C100); GPX4 (1:1000, Abcam, Ab125066); IL1b (1:500, Santa Cruz, CA, USA, sc-7884); and caspase 3 (1:500, Santa Cruz, sc-7272). Mouse monoclonal anti-α-tubulin antibody (1:10,000; reference T5168; Sigma-Aldrich) or anti-GAPDH (1:5000; reference mab374; EMD Millipore ) were used as the loading control.

### 2.3. Histology and Immunohistochemistry

Paraffin-embedded kidney sections were stained using standard histology procedures [45]. Periodic acid-Schiff (PAS, Sigma-Aldrich) stained slides were quantified, assessing tubular damage as tubular dilation and interstitial inflammatory infiltrate as arbitrary units, as previously described by [46]. Slides were quantified using Image Pro-plus Software (Rockville, MD, USA) to determine the positive red staining area relative to the total area.

Immunohistochemistry (IH) was performed in 3 μm thick tissue sections. The PTlink system (DAKO) was used to retrieve antigens using sodium citrate buffer (10 mM) adjusted to pH 6 or pH 9, depending on the requirements of the immunohistochemical marker (primary antibody used). Endogenous peroxidase was blocked. Sections were incubated for 1 h at room temperature with 1X Casein Solution (Vector Laboratories; Mowry Ave, Newark, CA, USA) to remove non-specific protein binding sites. Then, primary antibodies were incubated overnight at 4 °C. Specific HRP-conjugated (DAKO) or horse anti-rabbit or horse anti-mouse biotinylated secondary antibodies (Amersham Biosciences; Buckinghamshire, UK) were used. The latter was followed by avidin–biotin complex incubation (Vector Laboratories, Mowry Ave, Newark, CA, USA). The signal was developed with substrate solution and 3,3-diaminobenzidine as a chromogen (Abcam). Finally, the slides were counterstained with Carazzi’s hematoxylin (Richard Allan Scientific; Canton, MI, USA). The primary antibodies used were goat polyclonal anti-4-hydroxynonenal (HNE 1:1000, Ab46545, Abcam); rat monoclonal anti–F4/80 (1:50; reference MCAP497; Bio-Rad), and rabbit polyclonal anti-CD3 (ready to use; reference IS503; DAKO). Specificity was checked with the omission of primary antibodies. For quantification, Image-Pro Plus software (Rockville, MD, USA) was used to determine the stained area relative to the total area or to count positive staining manually in 4 randomly chosen fields (×200 magnification).

### 2.4. Gene Expression Studies

RNA samples from the renal cortex were isolated with TRItidy G^TM^ (PanReac Darmstadt., Germany). cDNA was synthesized using a High-Capacity cDNA Archive kit (Applied Biosystems, Waltham, MA, USA) using 2 μg total RNA following the manufacturer’s instructions. Quantitative gene expression analysis was performed using an AB7500 fast real-time PCR system (Applied Biosystems) using fluorogenic TaqMan MGB primers. The mouse assays IDs were: *Lcn2*: Mm01324470_m1, *Havcr1* (*Kim-1*): Mm00506686_m1, *Ccl-2*: Mm00441242_m1, *Hmox1*: Mm00516005_m1, *Nfe2l2* (*Nrf2*): Mm00477784_m1, *Catalase:* Mm00437992_m1; *Nlrp3* (Mm00840904_m1); *Gpx4* (Mm-04411498_m1); *Ripk3* (Mm00444947_m1); *Mlkl* (Mm012444219_m1); *Il-18* (Mm00434226_m1); *Il1b* (Mm00434228_m1); *Il-6* (Mm00446190_m1); and *Ccl-5* (Mm01302427_m1). Data were normalized to *Gapdh*: Mm99999915_g1 (Vic). The mRNA copy numbers were calculated for each sample with the instrument software using the Ct value (“arithmetic fit point analysis for the lightcycler”). The results were expressed in copy numbers, calculated relative to 3-month-old mice after normalization against *Gapdh.*

### 2.5. Statistical Analysis

The results are expressed as fold increases with respect to the average of 3-month-old mice as mean ± standard deviation (±SD). The normality of the sampling distribution was evaluated using the Shapiro–Wilk test. If the samples followed the Gaussian distribution, then a one-way ANOVA was used followed by the corresponding post hoc analyses. To compare non-parametric samples, a Kruskal–Wallis and a subsequent post hoc analysis was performed. GraphPad Prism 8.0 (GraphPad Software, San Diego, CA, USA) was used to develop graphics and statistical analysis. A *p*-value < 0.05 was considered statistically significant.

## 3. Results

### 3.1. CCN2 Deficiency Diminishes Kidney Damage and Inflammation in Murine AKI Induced with Folic Acid

To investigate the role of CCN2 in experimental AKI, nephrotoxicity was induced with folic acid (FA) administration in CCN2 conditionally deficient mice and corresponding control mice. 

The renal lesions were evaluated using PAS staining, scoring tubular dilatation, loss of/flattened brush border, and cast formation (Figure 1A,B). In the absence of CCN2, tubular damage induced with FA-AKI was slightly milder than in wild-type mice. The kidney damage biomarker KIM-1 was also evaluated. In FA-injected kidneys from wild-type mice, KIM-1 protein levels were markedly upregulated in injured tubules. KIM-1 immunostaining was reduced in CCN2-deficient mice with AKI (Figure 1A,C). Accordingly, in FA-AKI, the kidney gene expression of *Havcr1* and *Lipocalin 2*, encoding KIM-1 and NGAL, respectively, was increased in WT mice, and there was a trend towards milder in CCN2-deficient mice (Figure 1D).

Next, we determined the effect of CCN2 deficiency on the regulation of the inflammatory response in FA-AKI. There was a significant decrease in the number of infiltrating T lymphocytes (CD3+ cells) and macrophages (F4/80+ cells) in the kidneys of CCN2-deficient mice with FA-injected mice compared to wild-type FA mice (Figure 2A,B). The recruitment of inflammatory cells into the kidney is mediated by the local production of proinflammatory mediators. In FA-AKI kidneys, the kidney expression of *Ccl-2* and *Ccl-5* was upregulated, and there was a trend for lower expression in CCN2-deficient mice (Figure 2C).

One of the main features of FA-AKI is tubular cell death. In CCN2-deficient mice, a significant decrease in cell death was observed, as determined using deoxynucleotidyl transferase-mediated digoxigenin-deoxyuridine nick-end labeling (TUNEL) staining (Figure 3A). To determine the specific type of cell death, we analyzed caspase 3 protein levels, representing the main pathway linked to apoptotic cell death. FA-injected WT mice had increased kidney active caspase 3, and this was unchanged with CCN2-deficiency, suggesting that apoptosis is not regulated by CCN2 and that other types of cell death should be involved (Figure 3B).

### 3.2. CCN2 Modulates Necroptosis in Murine AKI Induced with Folic Acid

Necroptosis is an inflammatory form of cell death. Recent studies have described the involvement of the key regulators of necroptosis in the pathogenesis of experimental AKI [47]. In this study, we found that in FA-AKI, CCN2 deletion decreased the renal gene expression of the key components of the necroptosis pathway: *Ripk-3* and *Mlkl* (Figure 4A). FA kidneys presented increased MLKL protein levels compared to healthy kidneys, as observed using Western blot. Importantly, renal MLKL protein expression was significantly lower in CCN2-deficient FA-AKI (Figure 4B).

### 3.3. RIPK3 Is a Key Pathway in CCN2 Inflammatory Responses in the Kidney

Previous preclinical studies have demonstrated that inflammatory cell infiltration is one of the earliest characteristics of renal damage induced by CCN2 in mice [23]. To investigate the role of RIPK3 in CCN2-induced responses in the kidney, CCN2(IV), the recombinant C-terminal fragment of CCN2 with biological activity, was injected intraperitoneally into *Ripk3*-deficient mice, and the findings were compared to CCN2-injected WT mice. The immunohistochemical characterization of kidney infiltrating immune cells revealed that *Ripk3* deficiency significantly reduced the number of infiltrating monocytes/macrophages (F4/80+ cells) and T lymphocytes (CD3+ cells) in the kidney of CCN2-injected mice to levels similar to those of control mice (Figure 5A,B). CCN2 upregulated the renal gene expression of *Ccl-2*, *Ccl-5*, and *Il-6*, and this was significantly milder in *Ripk3*-deficient mice (Figure 5C), supporting an anti-inflammatory effect of *Ripk3* deficiency.

### 3.4. CCN2 Deficiency Decreased NLRP3 Inflammasome Components in Murine FA-AKI

Next, we investigated whether CCN2 deficiency could modulate NLRP3 inflammasome pathway activation in the kidney. The canonical NLRP3 inflammasome activation leads to caspase 1 activation and subsequent IL-1β or IL-18 maturation and release [5]. In AKI-FA, the NLRP3 inflammasome gene and protein levels were upregulated, whereas, in CCN2 deficiency, NLRP3 expression levels were similar to the controls (Figure 6A,B). Gene expression levels of *Il-18* and *Il-1β* were upregulated in FA-AKI compared to the controls. Moreover, in FA-AKI the immature and active forms of IL-1β in injured kidneys were increased. Importantly, CCN2 deficiency significantly decreased *Il-1β* gene expression (Figure 6A) and down-regulated renal protein levels of IL-1β (Figure 6C,D).

### 3.5. CCN2 Regulates NLRP3 Inflammasome Components in the Kidney

Next, we investigated whether CCN2 activates NLRP3 inflammasome components in the kidney under physiological conditions. In CCN2(IV)-injected mice, several components of the NLRP3 inflammasome pathway, such as *NLRP3, IL-18*, and *IL-1β* were upregulated in the kidney, compared to control mice, both at gene (Figure 7A) and protein (Figure 7B) expression levels. RIPK3 can promote cell death and/or inflammation through inflammasome activation [5]. Next, we evaluated the role of RIPK3 in NLRP3 inflammasome regulation. CCN2 administration to RIPK3-deficient mice resulted in the downregulation of kidney *NLRP3* gene and protein expression levels and of kidney *Il-1β* gene expression to control values (Figure 7A).

### 3.6. CCN2 Deficiency Modulates Lipid Peroxidation and the NRF2-Associated Antioxidant Response in Murine FA-AKI

Lipid peroxidation is a process that is present in renal damage impairing nephron function and promoting cell death [48]. To assess lipid peroxidation resulting from oxidative stress, kidney sections were stained for 4HNE. Lipid peroxidation was increased in FA-AKI mice (Figure 8A,B) and decreased in CCN2-deficient FA-AKI mice (Figure 8A,B). Glutathione peroxidase 4 (GPX4) reduces toxic lipid peroxidation. Kidney GPX4 gene and protein expression levels were reduced in FA-AKI mice, and those changes were restored in CCN2-deficient FA-AKI mice (Figure 8C,D). In addition, the regulation of antioxidant components, such as *Catalase,* and several NRF2-dependent genes, including *Hmox-1* by CCN2, was explored. In FA-AKI, increased kidney gene expression of *Nrf2* and *Hmox-1* as well as reduced levels of *Catalase* were observed (Figure 8E). CCN2 deficiency restored *Nrf2* and *Hmox-1* but not *Catalase* gene expression to control values (Figure 8E).

## 4. Discussion

In this study, for the first time, we characterized the key role of CCN2 in the activation of the RIPK3/NLRP3 pathway and its involvement in the regulation of the inflammatory and redox response in the kidney. These findings can be used to support further research on the therapeutic target potential of these pathways to modulate deleterious inflammation in the kidney and improve AKI outcomes.

CCN2, also known as CTGF in the past, is a matricellular protein that exerts pleiotropic actions involved in the regulation of relevant biological processes including cell proliferation, angiogenesis, migration, ECM remodeling, and cellular senescence [20,21,22,23,24,27,49]. Early studies demonstrated the role of CCN2 in kidney injury as a downstream mediator of profibrotic TGF-β1 [50] or angiotensin II [51,52], regulating ECM accumulation through the regulation of signaling pathways including the MAPK cascade and Smad [27]. Later studies extended the deleterious actions of CCN2 in the kidney via induction of an inflammatory response, and in some cases, this was related to the renal activation of the NF-κB pathway or via the modulation of the Th17 immune response [23,33]. CCN2 is a non-canonical EGFR ligand that directly binds to EGFR in kidney tubular cells, activating downstream signaling [49,53,54]. CCN2, both the complete molecule and its C-terminal module, can activate the EGFR pathway to promote proinflammatory and profibrotic responses in the kidney [49,53]. Previous studies using CCN2 knockout mice demonstrated that the key role of CCN2 in renal fibrosis was via the induction of cell cycle arrest in chronic FA nephropathy [31] and via the activation of cell senescence in IRI kidney damage [32]. These studies emphasize the relevance of CCN2 in renal diseases. In the present study, we described for the first time that CCN2 deficiency protected from FA-AKI by decreasing tubular cell injury, as shown by milder tubular dilatation, cast formation, loss of tubular brush border, and KIM-1 protein expression, a proposed biomarker for renal injury [55,56], as well as decreasing cell death. This result identifies a novel role of CCN2 in the regulation of tubular cell death induced by a nephrotoxic insult.

Apoptosis has been described in different experimental models of AKI, such as IRI or exposure to nephrotoxic compounds (including cisplatin or FA) [57,58,59,60]. Apoptosis can be assessed with caspase 3 activation and TUNEL staining for DNA damage [61]. We found that CCN2 deficiency reduced kidney TUNEL staining in FA-AKI. In tubular epithelial cells, caspase blockade can regulate different cell death types (from apoptosis to regulated necrosis pathways) induced by inflammatory cytokines [13,62]. CCN2 deficiency did not decrease kidney caspase 3 protein in FA-AKI, suggesting that the main CCN2-regulated form of cell death in FA-AKI is not apoptosis. Importantly, TUNEL positivity is not only limited to apoptosis: it can be detected in regulated necrosis cells [63]. Thus, another type of cell death, likely regulated necrosis, may be involved in CCN2-induced tubular cell death. Previous studies have described that regulated necrosis pathways are activated during FA-AKI and are associated with the upregulation of components in this cell death pathway, such as RIPK3 and MLKL, contributing to cell death and kidney damage [13,47]. Recent in vivo and in vitro studies suggest that necroptosis blockade protects from AKI induced with different insults [12,47,64,65,66,67]. Here, we propose that CCN2 can regulate cell death through the activation of necroptosis. In the absence of CCN2, FA-AKI-associated changes in *Mlkl* and *Rip3k* gene expression, as well as in MLKL protein levels, were no longer observed suggesting that modulation of necroptosis by CCN2 can induce cell death.

The activation of several inflammatory multiproteic complexes, like the NLRP3 inflammasome pathway, plays an important role in human and experimental AKI and CKD [68,69,70,71,72]. Preclinical studies in contrast-induced AKI reported that NLRP3 inflammasome activation and secretion of its downstream components IL-1β and IL18 exacerbated renal damage and apoptosis [73,74]. In addition, NLRP3 deficiency alleviated kidney injury through the attenuation of inflammation in the experimental metabolic syndrome [75]. In this study, we observed increased renal gene (*Nlrp3, Il-1β,* and *Il18*) and protein levels (NLRP3 and IL-1β) of inflammasome components in FA-AKI that were restored to control values in CCN2-deficient mice. Cell death and inflammasome activation may be interconnected [76,77]. MLKL deficiency or NLRP3 inhibition reduced kidney infarction in murine cholesterol crystal embolism [78]. In bone marrow-derived macrophages, MLKL-mediated membrane damage activated NLRP3 [79]. In murine IRI-AKI, the upregulated expression of RIPK3 and MLKL in renal proximal tubular cells induced necroptosis and contributed to NLRP3 inflammasome activation, and *Ripk3* or *Mlkl* deficiency reduced tubular cell necroptosis, macrophage infiltration, and NLRP3 inflammasome activation [80]. These results suggest that CCN2 can modulate cell death in experimental AKI through RIPK3/NLRP3 inflammasome activation and support future research targeting these pathways to treat AKI.

In some contexts, CCN2 can act as a proinflammatory mediator, including renal damage, promoting cellular infiltration by activating NF-κB [23]. In this study, we found that CCN2 deficiency reduced the inflammatory response caused by FA-AKI, diminishing T cell and macrophage recruitment and the gene expression of proinflammatory factors. Previously, CCN2 deletion also ameliorated inflammatory-related events in the acute phase of IRI [40]. These data extend previous studies describing the role of CCN2 in the regulation of sterile inflammation to pathological processes associated with nephrotoxic or ischemic insults causing AKI. In the present study, we described for the first time a novel mechanism by which CCN2 promotes kidney inflammation, the recruitment of NLRP3 inflammasome, and the activation of the RIPK3 signaling pathway in AKI. RIPK3 deficiency reduced the CCN2-induced kidney gene expression of NLRP3 inflammasome components including NLRP3, IL18, and IL-1β as well as the protein levels of NLRP3 in mice. Moreover, CCN2-induced kidney inflammation was prevented in RIPK3-deficient mice, indicating the key role of necroptosis in the inflammation induced by CCN2. Other findings support these observations. Pharmacological inhibition of RIPK3 is known to alleviate AKI by inhibiting necroptosis and inflammation [12,47,64,65,66,67,81]. In a recent study, specific targeting of bone marrow-derived RIPK3 resulted in milder NF-κB activation and milder kidney inflammation independently of tubular cell death [82]. NF-κB is one of the main proinflammatory pathways activated by CCN2 in the kidney [23]. These results support the need to develop novel RIPK3 inhibitors as anti-inflammatory drugs to treat kidney disease. However, the knowledge gap that must be addressed is understanding the conformational changes in RIPK3 that trigger apoptosis [19] to develop better RIPK3 pharmacological inhibitors.

CCN2 is a potent inducer of oxidative stress in several organs and cell types. CCN2 induces ROS production via the NOX-1/EGFR pathway in the murine aorta and in cultured vascular smooth muscle cells [39]. Furthermore, CCN2 also causes ROS accumulation in cultured fibroblasts [83]. However, in cultured cardiac myocytes, CCN2 overexpression protected against doxorubicin-induced oxidative stress and cell death [84], thus showing the opposite effect. In the kidney, CCN2 deficiency decreased oxidative stress induced by IRI in mice [40], as we also found here for FA-AKI, suggesting that CCN2 activation of oxidative stress may contribute to kidney injury. Some data also link RIPK3 to oxidative stress. RIPK3 upregulates NOX4 and promotes sepsis-induced AKI by inducing oxidative stress and mitochondrial dysfunction [85]. Kidney lipid peroxidation results from oxidative stress, i.e., an excess ROS generation with respect to antioxidant defenses, impairing nephron function and promoting cell death including apoptosis, autophagy, and ferroptosis [86,87,88,89]. CCN2 deficiency decreased kidney lipid peroxidation, as we showed by lower renal 4HNE levels. Glutathione peroxidase 4 (GPX4, phospholipid hydroperoxide glutathione peroxidase, PHGPx) reduces phospholipid hydroperoxides protecting them from toxic lipid peroxidation [90]. In this regard, kidney GPX4 gene and protein levels were also restored to control values in CCN2-deficient mice with FA-AKI. All these data suggest that CCN2, by the induction of oxidative responses, can contribute to kidney damage.

Nrf2 (nuclear factor erythroid 2-related factor 2) is a transcription factor deeply associated with protection against oxidative stress and the modulation of inflammation, which are key processes in the maintenance of cellular homeostasis [91]. The antioxidant response regulated by Nrf2 is driven by the modulation of genes encoding proteins related to the detoxification/elimination of pro-oxidative compounds, including H_2_O_2_ [92]. Many experimental studies have proven that NRF2-regulated antioxidant responses are activated in AKI [93,94,95,96,97]. Here, we showed that during FA-AKI, CCN2 deficiency regulated gene expression of the antioxidant factor catalase and restored changes in the Nrf2 pathway, including *Nrf2* and *Hmox-1* mRNA levels, the latter being a NRF2-controlled gene. In our previous studies on IRI-AKI, Nrf2 target genes *Hmox-1* and *Nqo1* were also modified in CCN2-deficient mice, which was linked to the modulation of cellular senescence [40]. All these results suggest that CCN2 can regulate redox processes associated with different detrimental mechanism in experimental AKI.

Some limitations should be acknowledged. This is a preclinical study that would require specific clinical studies to confirm its translational value. In this regard, the impact of complete gene deficiency, sometimes through life, may differ from the impact of drugs targeting specific mediators. Finally, systemic administration of CCN2 may not fully reproduce the microenvironmental conditions observed where CCN2 is being locally produced. Regarding RIKP3/NLRP3, better pharmacological interventions should be explored.

In conclusion, our preclinical data clearly show that CCN2 plays a key role in sterile inflammation and AKI by modulating the RIKP3/NLRP3/NRF2 inflammatory pathways. These data add novel information to prior experimental findings supporting the role of the RIKP3/NLRP3/NRF2 pathway in kidney inflammation. Our results identify CCN2 as a novel therapeutic target to modulate this pathway that should be explored further to design novel, more effective therapeutic approaches to kidney disease that reduce inflammation.

## Figures and Tables

**Figure 1 antioxidants-12-01541-f001:**
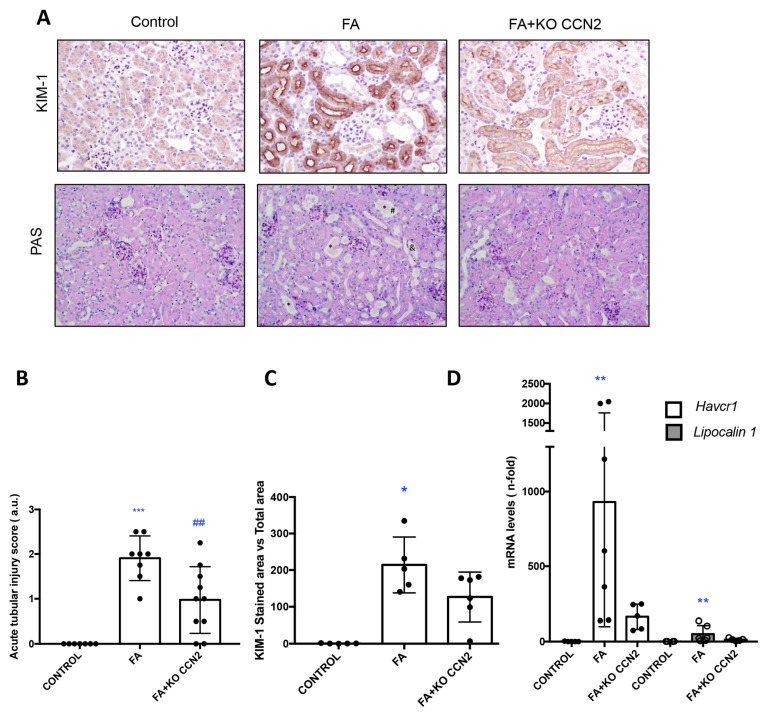
**CCN2 deficiency in mice diminished tubular damage and KIM-1 renal expression in acute kidney injury induced with folic acid administration**. Conditional CCN2-deficient mice (CCN2flox/floxROSA26-ERT/Cre) were injected with corn oil (vehicle) or with tamoxifen to induce CCN2 deletion. Three weeks later, CCN2 deletion was confirmed, and mice were divided into two additional groups: one received a single intraperitoneal injection of FA (250 mg/kg), and the other received a single intraperitoneal injection of vehicle (0.3 mol/L sodium bicarbonate); all mice were studied 48 h later. (**A**) Representative images of PAS staining that show the loss of the proximal tubule brush border, tubular dilatation, and intra-cast formation in animals exposed to FA compared with vehicle mice and representative images of Kim-1 staining. Magnification 200×. (**B**) Quantification of AKI score (addition of several damaged structures). (*, tubular cast; # and &, tubular brush border). (**C**) Quantification of KIM-1 expression in renal tissue. (**D**) Renal mRNA expression of *Havcr-1* and *Lipocalin2* were evaluated using RT-PCR. Data are expressed as mean ± SD of 4–8 animals per group. * *p* < 0.05; ** *p* < 0.01; *** *p* < 0.001 vs. control ## *p* < 0.001 vs. FA.

**Figure 2 antioxidants-12-01541-f002:**
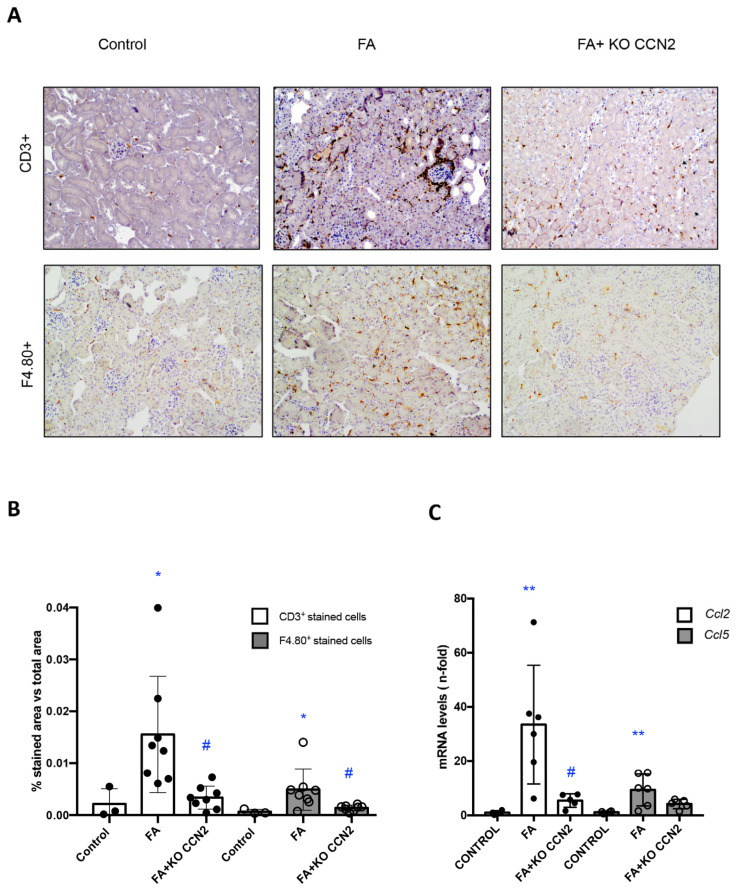
**CCN2 deletion in mice diminished inflammatory T cells and macrophages in the FA-AKI model**. (**A**) Representative immunohistochemistry (IH) images identifying inflammatory T cell (CD3+ T lymphocytes) and macrophage (F4/80+ monocytes/macrophages) infiltration in paraffin-embedded kidney sections. Magnification 200×. (**B**) CD3+ and F4/80+ IH staining quantification expressed as mean of stained area vs. total area ± SEM of 4–7 animals per group. (**C**) Kidney gene expression of *Ccl-2 (Mcp-1)* and *Ccl-5 (Rantes)* was evaluated using RT-PCR. Data are expressed as mean ± SD of 4–8 animals per group. * *p* < 0.05; ** *p* < 0.01 vs. control, # *p* < 0.05 vs. FA.

**Figure 3 antioxidants-12-01541-f003:**
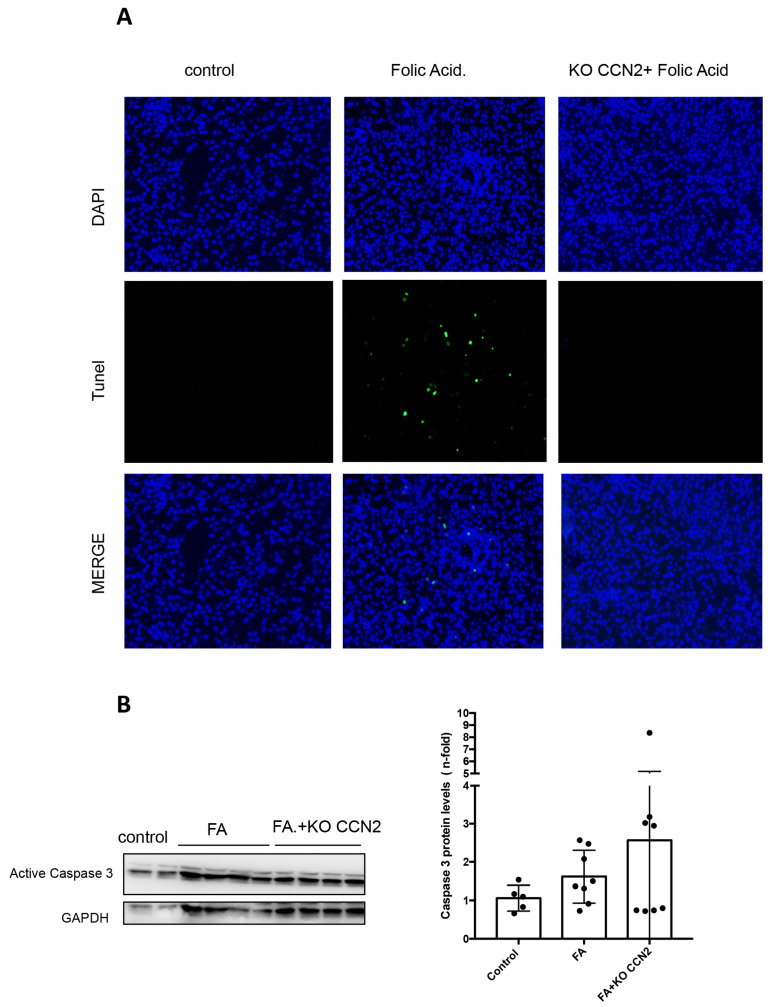
**CCN2 deficiency reduced cell death in murine AKI induced with folic acid**. (**A**) Cell death was assessed using TUNEL fluorescent staining in renal tissue. Magnification 200×. (**B**) Caspase 3 protein levels were evaluated using Western blot. Data are expressed as mean ± SD of 4–8 animals per group.

**Figure 4 antioxidants-12-01541-f004:**
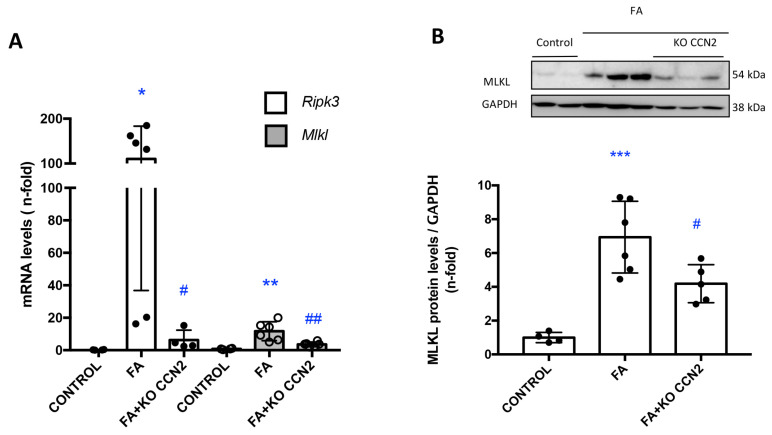
**CCN2 deficiency reduced necroptosis-related biomarkers in murine FA-AKI**. (**A**) The kidney gene expression of *Ripk-3* and *Mlkl* was evaluated using RT-PCR. (**B**) MLKL protein levels were assessed using Western blot. Data are expressed as mean ± SD of 4–8 animals per group. * *p* < 0.05; ** *p* < 0.01; *** *p* < 0.001 vs. control, # *p* < 0.05; ## *p* < 0.01 vs. FA.

**Figure 5 antioxidants-12-01541-f005:**
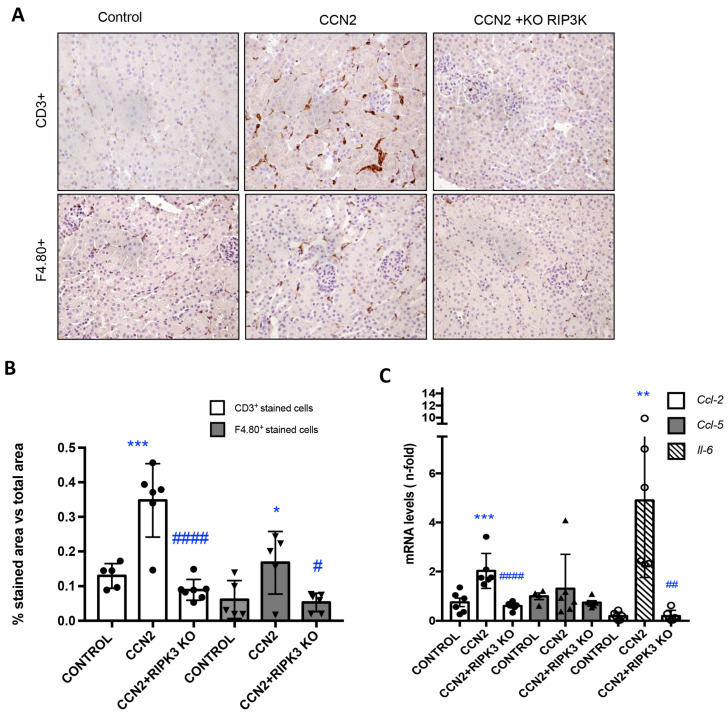
**RIPK3 deficiency diminished inflammatory cell infiltration and proinflammatory gene expression in mice following CCN2 administration**. Mice were injected with a single intraperitoneal dose of 2.5 ng/g of body weight recombinant CCN2(IV) and studied 24 h later. (**A**) Representative immunohistochemistry images identifying inflammatory T cell (CD3+) and macrophage (F4/80+) infiltration. Magnification 200×. (**B**) Immunohistochemistry quantification. (**C**) Kidney gene expression of *Ccl-2* (*Mcp-1*), *Il-6*, and *Ccl-5* (*Rantes*) was evaluated using RT-PCR. Data are expressed as mean ± SD of 4–8 animals per group. * *p* < 0.05; ** *p* < 0.01; *** *p* < 0.001 vs. control, # *p* < 0.05; ## *p* < 0.01; #### *p* < 0.0001 vs. CCN2.

**Figure 6 antioxidants-12-01541-f006:**
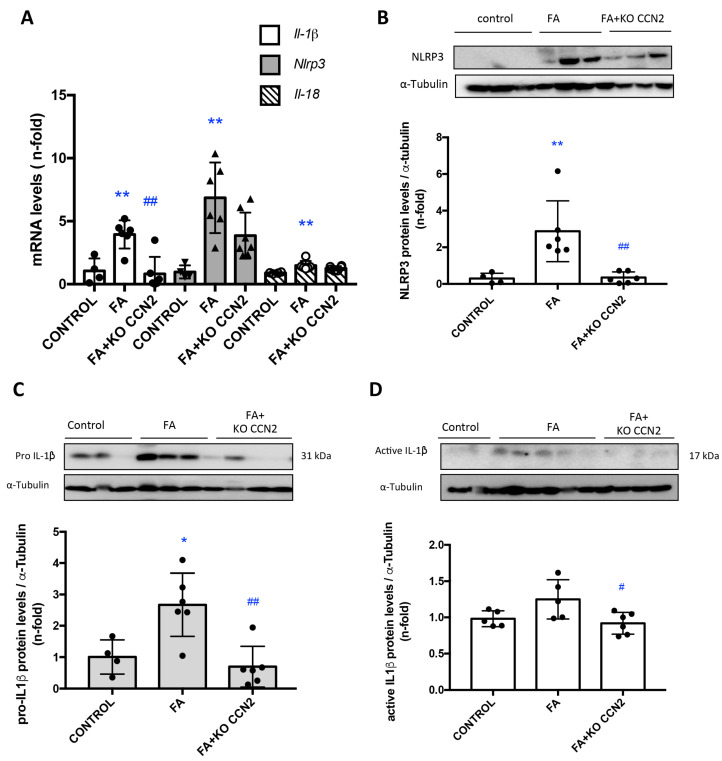
**CCN2 deficiency diminished the gene and protein expression of inflammasome components in a murine FA-AKI**. (**A**) Kidney gene expression of *Il-1b*, *Nlrp3*, and *Il-18* was evaluated using RT-PCR. (**B**) NLRP3, pro-IL-1β (**C**), and active IL-1β (**D**) protein levels were assessed using Western blot. Data are expressed as mean ± SD of 4–8 animals per group. * *p* < 0.05; ** *p* < 0.01; vs. control, # *p* < 0.05; ## *p* < 0.01 vs. FA.

**Figure 7 antioxidants-12-01541-f007:**
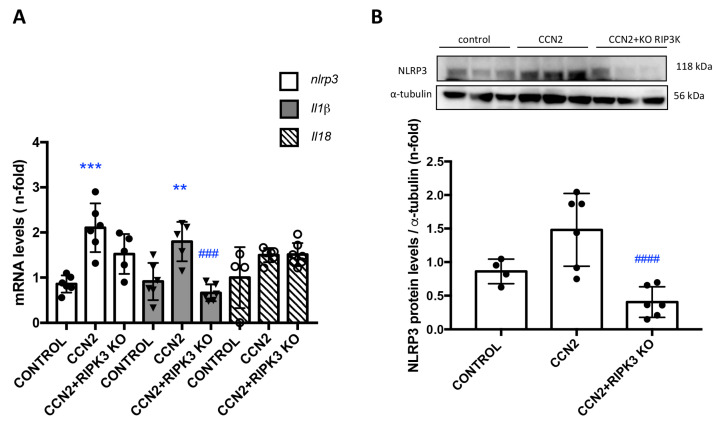
**RIPK3 deficiency diminished inflammasome components following CCN2 administration**. Mice were injected with a single intraperitoneal dose of 2.5 ng/g of body weight recombinant CCN2(IV) and studied 24 h later. (**A**) Kidney gene expression of *Il-1β*, *Nlrp3*, and *Il-18* was evaluated using RT-PCR. (**B**) NLRP3 protein levels were assessed using Western blot. Data are expressed as mean ± SD of 4–8 animals per group. ** *p* < 0.01; *** *p* < 0.001 vs. control, ### *p* < 0.001; #### *p* < 0.0001 vs. CCN2.

**Figure 8 antioxidants-12-01541-f008:**
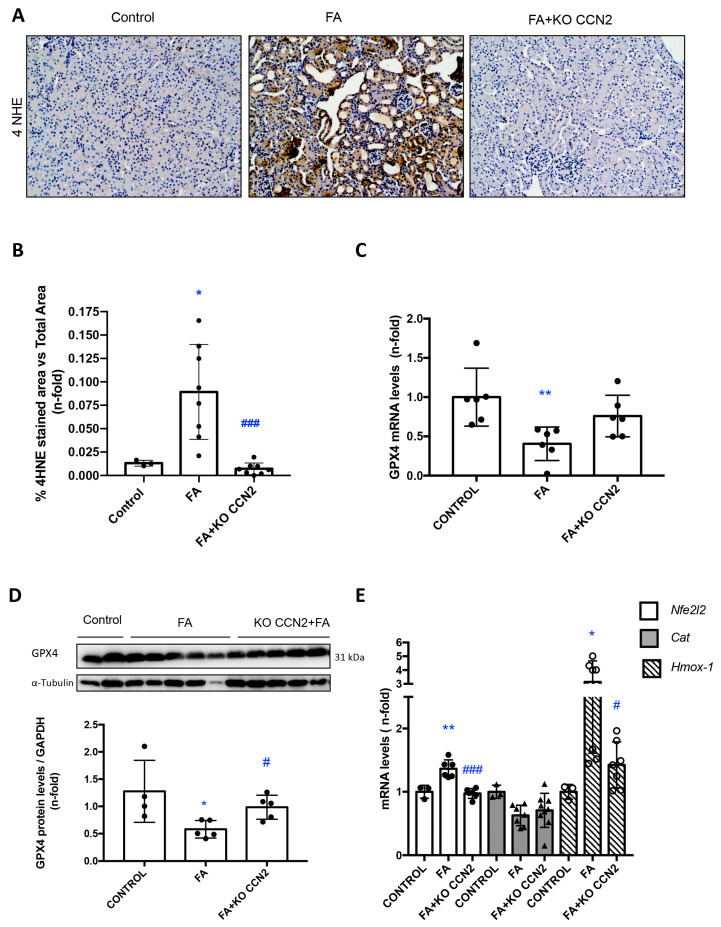
**CCN2 deficiency diminished lipid peroxidation and the gene expression of antioxidant components in murine FA-AKI**. Mice were injected with folic acid (FA) 300 mg/kg or vehicle (sodium bicarbonate 0.3 mol/L) and studied 48 h later. (**A**) Paraffin-embedded kidney sections were stained with an anti-4-HNE antibody. Representative immunohistochemistry images at 200× magnification. (**B**) Immunohistochemistry quantification. (**C**) Kidney GPX4 mRNA levels were studied using RT-PCR. (**D**) Kidney GPX4 protein levels were studied using Western blot. (**E**) Kidney gene expression of *Nfe2l2* which encodes Nrf2, *catalase* (Cat), and *Hmox-1* was evaluated using RT-PCR. Data are expressed as mean ± SD of 4–7 animals per group. * *p* < 0.05; ** *p* < 0.01; vs. Control, # *p* < 0.05; ### *p* < 0.001 vs. FA.

## Data Availability

No multi-omics data were obtained in this article. Results used in this publication will be shared upon reasonable request addressed to the corresponding author.

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
