# Peer review of "CCN2 Activates RIPK3, NLRP3 Inflammasome, and NRF2/Oxidative Pathways Linked to Kidney Inflammation"

_antioxidants, 2023, doi:10.3390/antiox12081541_

Round 1
Reviewer 1 Report
The manuscript is informative and well-written, presenting a proof-of-principle argument for CCN2 inhibition as a possible therapeutic pathway. The authors should present data as mean +/- s.d. rather than SEM, these are not theoretical distributions but actual observations made on a sample. Also, please add a study limitations paragraph.
Author Response
Answer to reviews
Reviewer 1
The manuscript is informative and well-written, presenting a proof-of-principle argument for CCN2 inhibition as a possible therapeutic pathway. The authors should present data as mean +/- s.d. rather than SEM, these are not theoretical distributions but actual observations made on a sample. Also, please add a study limitations paragraph.
As suggested figures have been changed to mean+/-SD. We have included a study limitation paragraph.
Reviewer 2 Report
This reviewer do not understand the purpose of the information provided on lines 43-74. these lines just provide a literature review and convey nothing important in terms of gap in the knowledge.
Gender of the mice is not clear in section 2.1. also provide justification if mice of one gender is selected. justify if there is mixture of the genders and how this was resolved in terms of statisitics.
authors should discuss limitations of the studies in light of existing studies
provide exact pH of the sodium citrate buffer. usually it is a 10mM sodium citrate pH6.0. range is not acceptable.
provide the species of the primary and secondary antibodies on lines 156-171.
arrange all bar charts in one line just like figure 1A.
bring the legend within the figure 4A.
remove line between fig5 panels and legend. explain what was compared to show p as # and *. fig 5c, asterisk are cropped.
what critical information is provided by section 3.4
line 397 fix IL1b.
The discussion reads like a literature review. it is not clear in the discussion what gap in knowledge this paper addresses that have not been looked at in past research.
Conclusion is non-existant to weak.
Author Response
Answer to reviews
Reviewer 2
This reviewer do not understand the purpose of the information provided on lines 43-74. these lines just provide a literature review and convey nothing important in terms of gap in the knowledge.
Introduction has been rewritten. In these lines provide important information to contextualize the inflammatory process inside of the renal damage and pathology, including information about the key role of necroptosis and NLRP3 inflammasome activation. Now, additional information has been included remarkinggaps in the knowledge in this area.
Gender of the mice is not clear in section 2.1. also provide justification if mice of one gender is selected. justify if there is mixture of the genders and how this was resolved in terms of statisitics.
Now, we have included gender and explanation (we have chosen one gender, male mice based on previous published data). No mixture was done.
authors should discuss limitations of the studies in light of existing studies
We have included a study limitation paragraph at the end of the manuscript.
provide exact pH of the sodium citrate buffer. usually it is a 10mM sodium citrate pH6.0. range is not acceptable.
Sorry for the mistake. This is not a range, we used pH6 or pH9 depends of the primary antibody requirements. We have rewritten the sentence for a correct understanding.
provide the species of the primary and secondary antibodies on lines 156-171.
Done
arrange all bar charts in one line just like figure 1A.
OK, we have disposed all the graphs in a line in figure 1A
bring the legend within the figure 4A.
Done
remove line between fig5 panels and legend. explain what was compared to show p as # and *. fig 5c, asterisk are cropped.
Ok, we have corrected the layout errors. In the case of the explanation of the use of p as # and * in this figure, you have the explanation at the end of the Figure 5 footnote.
what critical information is provided by section 3.4
The section 3.4 of the results indicates that necroptosis process is key signalling pathway in the proinflammatory response induced by CCN2 in the kidney. This a novel data as explained in discussion.
line 397 fix IL1b.
In this line the cytokine is written different due to the HUGO gene nomenclature that indicate if you refer to the murine gene expression you need to uselowercase and italics. We used these nomenclature in all the sections of the manuscript and the figures.
The discussion reads like a literature review. it is not clear in the discussion what gap in knowledge this paper addresses that have not been looked at in past research.
We have rewriten the discussion, including previous papers and the novelty of our findings, as well as futurre perpectives in research.
Conclusion is non-existant to weak.
Now a conclusion paragraph has been included.
Round 2
Reviewer 2 Report
Introduction is more bloated now, with lines 46-72 is still a collection of literature (~15 references), which do not lead to the gaps in the knowledge. The only lines that make the introduction clear and easy to read are to move line65 “More….cells” and line 70 “however…. Injuries”. These two lines are the only important ones that need to be added to para between lines 74-85.
Also why the introduction restarts for CCN2 again at line 86. The key sentence here is line 89 “Early… regulation” and line 97 “moreover… diseases”
If your question is about NLRP3/RIPK3 pathway, why are the reviewer and reader punished for reading this disjointed introduction? Is it because the authors have gone through 60 papers just to write the introduction? The para on lines 74-85 is a strong para in this intro, where the key parts should be presented and end with line 103 about the work you want to do. Using 60 references to write an introduction suggests the authors do not understand the work done. This introduction needs to be precise to the point.
Thanks for providing the gender of the mice used. However the gender and the justification should have been done in the section for animals on line 118 onwards. Species of mice and their age should be part of the section “animals” instead scattered all over the methods. If you are using the mice, you should also mention how many mice were used in the experiments in the methods section. At the same time the number of mice used should also be mentioned in the figure legends for reviewer. See lines 137-143. What was the background for CCN2 mice? Generation of CCN@ mice should be part of “animals” section.
50ug protein per well of a SDS-PAGE is overloading when your alpha-tubulin band is super saturated your densitometric analysis is going to be unreliable.
Line 185, the pH is requirement for the antibody as stated in the response, but you are using histochemical marker in the text. This is the reason 99% of the biomedical research is non reproducible.
Author mentions “several limitations…” on line 639. There are only two listed. Discuss in light of information on this paper. https://onlinelibrary.wiley.com/doi/10.1111/j.1365-2613.2012.00845.x
Author Response
Comments and Suggestions for Authors
1) Introduction is more bloated now, with lines 46-72 is still a collection of literature (~15 references), which do not lead to the gaps in the knowledge. The only lines that make the introduction clear and easy to read are to move line65 “More….cells” and line 70 “however…. Injuries”. These two lines are the only important ones that need to be added to para between lines 74-85.
We reduced as much as possible the content from lines 46-72 as reviewer suggests.
2) Also why the introduction restarts for CCN2 again at line 86. The key sentence here is line 89 “Early… regulation” and line 97 “moreover… diseases”
We have reduced this paragraph trying to focus in only the key information to the reader understands the context of the study.
3) If your question is about NLRP3/RIPK3 pathway, why are the reviewer and reader punished for reading this disjointed introduction? Is it because the authors have gone through 60 papers just to write the introduction? The para on lines 74-85 is a strong para in this intro, where the key parts should be presented and end with line 103 about the work you want to do. Using 60 references to write an introduction suggests the authors do not understand the work done. This introduction needs to be precise to the point.
We reduced as much as possible the content of the introduction as reviewer suggest. However, in the introduction we have followed the same structure of abstract, including a brief explanation of the importance of renal damage in our society and the lack of treatment in this field. Next, we have explained the proposed mechanisms involved in the inflammatory process in the kidney (inflammasome), with special attention to AKI-related processes (necroinflammation), and finally information about CCN2. Therefore, we consider that the actual introduction covers all the topics investigated in this paper. In particular, we believe that the inflammasome and its action mechanism as well as its role in renal damage have to be explained to the reader (who may not know). Regarding references, as good practices we have included original sources, and only very recent review papers. In this Journal there is no reference number limit.
4) Thanks for providing the gender of the mice used. However the gender and the justification should have been done in the section for animals on line 118 onwards. Species of mice and their age should be part of the section “animals” instead scattered all over the methods. If you are using the mice, you should also mention how many mice were used in the experiments in the methods section. At the same time the number of mice used should also be mentioned in the figure legends for reviewer. See lines 137-143. What was the background for CCN2 mice? Generation of CCN@ mice should be part of “animals” section.
The “animal section” in this manuscript (section 2.1) included two sections inside about the different experimental models of renal damage that we developed in the manuscript (Folic acid nephropathy and CCN2 administration). The gender justification would be included in each of them because the justification is different and it is not possible to include as a generic form in the beginning of the section 2.1. On the other hand we include the number of mice of each experimental group more clearly in this part of the manuscript and as reviewer can see in the footnotes of each figure, the information about the number of animals it was already included. We included the background of CCN2 KO mice and the generation of these animals in a new subsection ( 2.1.1 ) in animals section ( 2.1).
5) 50ug protein per well of a SDS-PAGE is overloading when your alpha-tubulin band is super saturated your densitometric analysis is going to be unreliable.
The quantity of protein used in the western blot is determined by the target protein of study and also by the efficiency and quality of the primary antibody that detect this protein of interest not by the protein that you used as loading control. In addition for a correct strictness of the results the loading control must hybridize in the same western blot membrane in which we previously hybridized and reveal for target protein of study. If we reduce the quantity of protein loaded in the gel for example in the case of active IL1β for a perfect visualization of α-tubulin (loading control); we guarantee the reviewer that we only will observe the α-tubulin but in no case a detectable IL1β protein levels. It is due to the reduced levels of this cytokine in the samples. In addition, is important to remark that the reviewer did not include this comment in the previous round of review.
6) Line 185, the pH is requirement for the antibody as stated in the response, but you are using histochemical marker in the text. This is the reason 99% of the biomedical research is non reproducible.
“Immunohistochemical marker” is a generic form to refer to a wide range of primary antibodies that we used in the manuscript. As referee has recommended we have rewritten the sentence to avoid concept errors.
7) Author mentions “several limitations…” on line 639. There are only two listed. Discuss in light of information on this paper. https://onlinelibrary.wiley.com/doi/10.1111/j.1365-2613.2012.00845.x
Thant you for your suggestion, we have corrected “several limitations…” for “ some limitations…”. In relationship with the discussion, we have included some references of this review manuscript that the reviewer suggest in the introduction and discursion, that including previous studies of our group from 2009 but is necessary to take account that this review is more focused in inflammation and fibrosis in chronic kidney disease and not in Acute kidney injury that is the main damage that happens in FA damage and is more important to focus the discussion in that way.